# OpenReview forum: "New News: System-2 Fine-tuning for Robust Integration of New Knowledge"
_ICLR.cc/2026/Conference — ICLR 2026 Conference Withdrawn Submission_

### Official Review · Reviewer_7vNx · 2025-10-27

**Soundness:** 2
**Presentation:** 2
**Contribution:** 2
**Rating:** 2
**Confidence:** 3

**Summary:**

This paper studies how to integrate new information into LLMs, aiming to bridge the performance gap between in-context learning (ICL) and naive fine-tuning (FT). The authors introduce a new dataset called "New News" to evaluate this capability. They propose a data generation protocol, termed "System-2 Fine-tuning" (Sys2-FT), which uses the model itself to generate paraphrases, implications, and question-answer pairs from the new information. This generated data is then used to fine-tune the model. The experiments show that these methods, particularly Self-QA, improve the model's ability to internalize and apply the new knowledge compared to naive fine-tuning.

**Strengths:**

* The paper studies a critical challenge in LLM research: how to update a model's internal knowledge base effectively without expensive retraining. Moving beyond in-context learning for new facts is an important research direction.

* Novel Dataset Contribution: The curation of the "New News" dataset is a great contribution. It is specifically designed to measure the internalization of hypothetical, non-trivial knowledge and its downstream consequences. Thus, the dataset is useful for future research in knowledge editing and continual learning.

**Weaknesses:**

* The central proposal, "System-2 Fine-tuning," appears to be a rebranding of existing, well-established techniques.
  - The "paraphrase" protocol is a straightforward form of data augmentation. Using a model to generate varied expressions of the same fact is not a novel technique.
  - The "Self-QA" and "implication" protocols are variations of self-instruction and context distillation, where a model generates its own training data (e.g., questions and answers) to "distill" knowledge. While effective, the paper does not differentiate its approach from this large body of prior work.

* The term "System-2 Fine-tuning", though is borrowed from cognitive science, is essentially a data augmentation strategy. The method does not change the model's reasoning process but rather rehearses variations of a fact during training. The framing of it as a new mode of "fine-tuning" is an overstatement; the fine-tuning itself is standard. The core idea is simply to generate more varied data.

* The paper presents the "contextual shadowing effect" as a surprising discovery. However, this is a direct result of the inconsistency between the training and evaluation conditions. During training with the context prefix, the model is optimized for an "open-book" task where it simply has to perform retrieval from the context. The evaluation is a "closed-book" task testing for knowledge internalization. This train/test discrepancy makes the resulting performance degradation an expected outcome.

**Questions:**

* The explanation of the data construction for fine-tuning is convoluted. Could the authors provide a concrete example that clearly shows the input for the paraphrase generation step, the generated paraphrases, and the final conversational data that is fed to the model?

* The novelty of the Sys2-FT protocols appears limited when viewed as a form of self-instruction or data augmentation. Can the authors more clearly articulate the contribution?

* Regarding the "contextual shadowing effect," given that the effect is a result of a train-test mismatch, what is the non-trivial takeaway?

---

### Official Review · Reviewer_PCVF · 2025-10-30

**Soundness:** 3
**Presentation:** 4
**Contribution:** 3
**Rating:** 6
**Confidence:** 4

**Summary:**

The paper proposes New News, a small but focused benchmark (75 hypothetical yet plausible news items; 375 downstream QA across math, coding, discoveries, leaderboards, events) to study in-weight integration of new information. As naive FT underperforms ICL, the paper then proposes System-2 Fine-Tuning (Sys2-FT): synthetic data generation from the news via Paraphrase, Implication, and Self-QA protocols, then SFT on the generated “replay” data. On the Qwen-2.5 family, Self-QA consistently narrows the FT–ICL gap, while largely preserving general capabilities (MMLU-Pro, GSM8K, IFEval). In additional analysis, the paper reveals two interesting phenomena : Contextual Shadowing (prefixing the news text before rephrases/QAs hurts learning the knowledge) and a Curse of Overexposure (FT can degrade ICL performance on the same news). The authors also present preliminary compute-normalized scaling suggesting larger models are more data-efficient than smaller models under Sys2-FT.

**Strengths:**

- The **New News** benchmark is manually constructed and sufficiently high quality to study knowledge integration via controlled experiments.
- Experiments are well designed and well motivated: FT under multiple protocols is compared against ICL, and **Self-QA** emerges as particularly effective.
- The analyses of **Contextual Shadowing** (prefixing news hurting learning) and **Overexposure** (FT degrading ICL on the same news) provide valuable insight into how different *views* of data affect learning.
- Preliminary results on knowledge retention, model-size effects for data generation/training, and compute scaling are clearly presented and informative.
- Writing is clear; the experimental setup and motivations are easy to follow.

**Weaknesses:**

- The small benchmark size (75/375) limits statistical power, domain coverage, and robustness of the reported effects.
- All results use the **Qwen-2.5** family; it remains unclear whether the observed phenomena (Contextual Shadowing, Overexposure) or the superiority of Self-QA generalize to other model families.
- The superiority of Self-QA may be overstated relative to paraphrase/implication, since training and evaluation both use QA formats, potentially aligning the conditional distribution in favor of Self-QA.
- It is unclear whether different FT protocols (Naïve/Paraphrase/Implication/Self-QA) are matched on **compute** (e.g., FLOPs/token parity), as some protocols naturally yield more tokens.
- While the experiments are careful, **Self-QA / System-2 FT** appears incremental conceptually, with prior work rephrasing raw data into QA (citations noted).

@article{maini2024rephrasing,
  title={Rephrasing the web: A recipe for compute and data-efficient language modeling},
  author={Maini, Pratyush and Seto, Skyler and Bai, He and Grangier, David and Zhang, Yizhe and Jaitly, Navdeep},
  journal={arXiv preprint arXiv:2401.16380},
  year={2024}
}

**Questions:**

1. Are FT protocols matched on effective compute (FLOPs), and if so, how do you normalize for token count differences across Naive/Paraphrase/Implication/Self-QA?
2. Do the Self-QA gains and the Shadowing/Overexposure phenomena persist on a non-Qwen family (e.g., Llama, Mistral), and at different scales?
3. How much do Self-QA’s improvements depend on evaluation being QA-style? Have you tested non-QA evaluation, or included an instruction-tuning stage after System-2 FT to normalize task formats across all protocols?

---

### Official Review · Reviewer_u2wP · 2025-10-31

**Soundness:** 2
**Presentation:** 2
**Contribution:** 2
**Rating:** 2
**Confidence:** 4

**Summary:**

This paper introduces New News, a dataset of 75 hypothetical news items across 5 domains (math, coding, discoveries, leaderboards, events) with downstream evaluation questions, designed to test whether LLMs can integrate new knowledge via finetuning. The authors demonstrate a substantial "FT-ICL gap", where models perform well when news is given in context but poorly after naive finetuning. To address this, they propose System-2 Finetuning (Sys2-FT), which generates synthetic training data (paraphrases, implications, or QA pairs) using the model's in-context capabilities. Their Self-QA protocol significantly outperforms naive finetuning, sometimes matching ICL performance. Key findings include: (1) the contextual shadowing effect (training with news in context degrades learning), (2) the curse of overexposure (finetuning can harm ICL ability), and (3) preliminary evidence of compute-based scaling laws for knowledge integration.

**Strengths:**

1. The proposed dataset, New News, could potentially provide a controlled testbed explicitly designed for knowledge integration evaluation.
2. While context distillation challenges are known, the systematic demonstration that prefixing news before training data catastrophically degrades learning is a clear empirical contribution with practical implications.
3. Very good visualization. Figures effectively communicate key findings (e.g., Fig. 2's FT-ICL gap, Fig. 4's protocol comparison, Fig. 10's scaling curves).
4. Knowledge integration via fine-tuning is an emerging, important topic in the field, where this work would have a practical impact.

**Weaknesses:**

1. The core contribution of synthetic data augmentation for finetuning is not sufficiently novel. Several recent works have explored similar or more sophisticated approaches: Learning Facts at Scale with Active Reading (Lin et al., 2025) and Cartridges: Lightweight and general-purpose long context representations via self-study (Eyuboglu et al., 2025). The active reading work actually compared to synthetic QA generation as one of their baseline.
2. The motivation for New News remains unclear, given existing benchmarks on continual learning and knowledge integration (ArchivalQA, StreamQA, LongHealth, SimpleQA, MTOB, etc.). The paper needs to explicitly answer: (1) What specific limitations of existing benchmarks does New News address? (2) What unique characteristics make it necessary (e.g., temporal aspects, knowledge types, difficulty)? (3) How does it enable analysis impossible with prior benchmarks?
3. For benchmark-side contributions, evaluating only the naive finetuning and the 3 "system-2" finetuning methods is not sufficient. As mentioned in W1, other advanced continual learning methods need to be evaluated as well. There are many categories of methods, such as finetuning (Lin et al., 2025; Eyuboglu et al., 2025), memory-augmented LLM (Tack et al, 2024; Berton et al., 2025), and test-time learning (Hu et al, 2023; Chen et al., 2025)
4. The use of "System-2" from dual process theory needs further justification. System-2 refers to slow, deliberate, conscious reasoning in humans. The paper mainly generates synthetic training data through prompted model outputs, a data augmentation approach, not System-2 reasoning. The model generating paraphrases or questions is still performing fast, automatic, attention-based (System-1) generation. More accurate framing would be self-play or simply self-augmentation. The authors should either (a) provide strong theoretical and empirical evidence that your prompted generation exhibits System-2 characteristics (e.g., show deliberation, error correction, multi-step reasoning traces), or (b) reframe the contribution more accurately as a data augmentation approach.
5. The paper jumps between Sys2-FT protocols (Sec. 4), contextual shadowing (Sec. 5), capability preservation (Sec. 6), cross-model analysis (Sec. 7), and scaling laws (Sec. 8) without a cohesive story. The authors may consider restructuring around a central question: "What determines successful knowledge integration via finetuning?"

### References
[1]  Hu, N., Mitchell, E., Manning, C. D., & Finn, C. (2023). Meta-learning online adaptation of language models. arXiv preprint arXiv:2305.15076.

[2] Tack, J., Kim, J., Mitchell, E., Shin, J., Teh, Y. W., & Schwarz, J. R. (2024). Online adaptation of language models with a memory of amortized contexts. Advances in Neural Information Processing Systems, 37, 130109-130135.

[3] Eyuboglu, S., Ehrlich, R., Arora, S., Guha, N., Zinsley, D., Liu, E., ... & Re, C. (2025). Cartridges: Lightweight and general-purpose long context representations via self-study. arXiv preprint arXiv:2506.06266.

[4] Chen, Z., Romanou, A., Weiss, G., & Bosselut, A. (2025). PERK: Long-Context Reasoning as Parameter-Efficient Test-Time Learning. arXiv preprint arXiv:2507.06415.

[5] Lin, J., Berges, V. P., Chen, X., Yih, W. T., Ghosh, G., & Oğuz, B. (2025). Learning facts at scale with active reading. arXiv preprint arXiv:2508.09494.

[6] Berton, G., Unnikrishnan, J., Tran, S., & Shah, M. (2025). CompLLM: Compression for Long Context Q&A. arXiv preprint arXiv:2509.19228.

**Questions:**

1. The paper mentions that news should be "simple yet have non-trivial downstream implications" (p.4). Can the authors provide more specific criteria? How did you ensure balance across difficulty levels? Were any news items tested and rejected, and why?
2. How did you ensure the 5 questions per news item have comparable difficulty? Are they all intended to be equally hard, or is there a difficulty gradient?
3. The implication protocol asks for "implications/consequences." Are these truly implications (logical consequences) or more like restatements emphasizing different aspects?

---

### Official Review · Reviewer_BXAi · 2025-11-01

**Soundness:** 3
**Presentation:** 2
**Contribution:** 2
**Rating:** 4
**Confidence:** 4

**Summary:**

This paper presented a dataset of hypothetical yet plausible news items named New News, where each sample is accompanied with downstream multiple-choice questions that require internalization of the news to answer correctly. This work investigated and discovered a performance gap between naive fine-tuning (FT) and in-context learning (ICL), and proposed the System-2 Fine-Tuning (Sys2-FT), which is a self-play data generation including paraphrases, implications, and Self-QA to distill contextual knowledge into model weights. Experimental results illustrated that Self-QA is the most effective protocol among them, the authors also observed a “contextual shadowing effect” where prefixing news during training degrades learning, thereby damaging the ICL performance.

**Strengths:**

1. The presented New News dataset is well-motivated and provides an insight for evaluating LLMs knowledge internalization beyond previous simple factual edits.

2. The Self-QA protocol of the Sys2-FT exhibited consistent improvement over naive SFT on injecting new knowledge to LLMs.

**Weaknesses:**

1. The New News dataset contains only 75 news articles and 375 questions, which is relatively small, and the count of news articles in each domain is limited, making it to evaluate model performance in broader scenarios comprehensively.

2. All experiments in this paper were based on the Qwen 2.5 models across different scales, however, other LLMs families, such as Llama 3 and Llama 3.x, which are not covered, which limits the generalizability of the conclusions and findings in this work.

3. This work focused on Qwen2.5-xB-Instruct models, and without analysis or discussion about reasoning LLMs that can perform better than Instruct LLMs in multiple downstream tasks, which undermines the generalizability of the proposed Sys2-FT.

4. While the authors observed the context shadowing effect and the curse of overexposure, this work failed to delve into the underlying mechanisms, lacking theoretical explanations and hurting the reliability of the Sys2-FT to real-world application.

5. The proposed Sys2-FT requires generating a large amount of self-generated data during fine-tuning, which can bring quite high computational budgets, and it would get worse for large-scale models and datasets. However, this paper didn’t’ adequately discuss the scalability of the Sys2-FT method to larger datasets, nor does it address how to address computational inefficiency.

**Questions:**

1. Why does the _contextual shadowing_ occur only when the news is prefixed? What about the common “Now I have got a problem about …” in common reasoning LLMs, which is also a prefix?

2. The current popular knowledge injection paradigm is continual pre-training (CPT), have the authors compared the performance or investigated the connections between the proposed System2-Finetuning and CPT?

3. How sensitive is the LLM performance to the self-play sample scales (1024 per news)?

4. The 1st citation for Figure 2 is at the bottom at Page 4, while the Figure 2 is at the top of Page 2, which is inconvenient for reading, please consider to reorganize the position of Figure 2.

---

### Note · Authors · 2025-11-20

I have read and agree with the venue's withdrawal policy on behalf of myself and my co-authors.